# Optical Fiber Sensors Based on Microstructured Optical Fibers to Detect Gases and Volatile Organic Compounds—A Review

**DOI:** 10.3390/s20092555

**Published:** 2020-04-30

**Authors:** Diego Lopez-Torres, Cesar Elosua, Francisco J. Arregui

**Affiliations:** 1Optical Communications Group, Electric and Electronic Engineering Department, Public University of Navarra, Edif. Los Tejos, Campus Arrosadía, 31006 Pamplona, Spain; cesar.elosua@unavarra.es (C.E.); parregui@unavarra.es (F.J.A.); 2Institute of Smart Cities (ISC), Centro Jerónimo de Ayanz, Campus Arrosadia, 31006 Pamplona, Spain

**Keywords:** microstructured optical fibers, solid core photonic crystal fibers, hollow core photonic crystal fibers, suspended core microstructured optical fibers, gasses, humidity and volatile organic compounds

## Abstract

Since the first publications related to microstructured optical fibers (MOFs), the development of optical fiber sensors (OFS) based on them has attracted the interest of many research groups because of the market niches that can take advantage of their specific features. Due to their unique structure based on a certain distribution of air holes, MOFs are especially useful for sensing applications: on one hand, the increased coupling of guided modes into the cladding or the holes enhances significantly the interaction with sensing films deposited there; on the other hand, MOF air holes enhance the direct interaction between the light and the analytes that get into in these cavities. Consequently, the sensitivity when detecting liquids, gasses or volatile organic compounds (VOCs) is significantly improved. This paper is focused on the reported sensors that have been developed with MOFs which are applied to detection of gases and VOCs, highlighting the advantages that this type of fiber offers.

## 1. Introduction

Optical fiber sensors (OFS) show diverse applications in fields of industry, environment and chemistry [1,2,3]. By monitoring the variations in optical power, shifts in phase, wavelength or polarization state, OFS can be used to measure physical or chemical magnitudes such as temperature, curvature, displacement, pressure, refractive index, electric and magnetic field, relative humidity (RH) and gasses, among others. Compared to their electronic counterparts, OFS also offer numerous advantages [4,5,6,7]. Moreover, OFS are electromagnetically inert and immune to electromagnetic interference, lightweight, their transmission losses are much lower than metallic wires and they can be wavelength multiplexed [8].

Thanks to the efforts exerted by numerous research groups since the 1970s [9,10], microstructured optical fibers (MOFs) were conceived and the development of OFS experienced a substantial evolution in terms of sensitivity, kinetics and selectivity. The main difference between MOFs and standard fibers is based on their structure: standard communications (single-mode fibers (SMFs) and multi-mode fibers (MMFs)) show a doped solid-core enclosed by a solid-cladding. This configuration implies a refractive index difference between the core and the cladding that enables the light guidance through the total internal reflection (TIR) mechanism. On the contrary, MOFs have air holes distributed along the fiber structure showing different dimensions and patterns. This distribution determines specific propagation modes as well as an effective refractive index. In this manner, modes can be confined in the MOF core or forced to travel through the cladding or the air holes. This possibility of increasing the evanescent field is a remarkable feature that standard fibers do not offer, and it is very relevant when preparing OFS.

MOF structures show unprecedented properties that lead to different types of propagation effects as well as propagation parameters [11,12]: small mode areas that enable soliton propagation whereas large mode areas enable to handle high-power optical beams; moreover, birefringence or group velocity dispersion (GVD) can be also controlled [13,14,15]. Based on these characteristics and considering their high versatility, several papers have demonstrated that MOFs are a good alternative to develop sensors to measure some physical and chemical parameters such as: curvature [16], strain [17], temperature [18], electric and magnetic field [19,20], pressure [21], torsion [22], refractive index [23], vibration and DNA [24], among others.

Not only silica is used to prepare MOFs [9] but also plastic is employed to manufacture microstructured polymer optical fibers (mPOFs) [25]. Methyl methacrylate (PMMA) is commonly the material which mPOFs are made of. Sensors for humidity [26] and strain [27] measurements based on mPOF are reported in the bibliography. The most relevant advantages of this kind of fibers are low Young’s modulus, ease of handling, biological compatibility and flexibility even for large diameters such as 1 mm [28]. The wavelength working range includes visible and near infrared [29] and the tolerance of the connectors is increased because of the large diameters. On the contrary, the main drawback is the transmission losses compared to MOFs, so that they are limited to short distance applications. Another disadvantage is their poor stability to temperatures above 160 °C (PMMA fusion point) [30,31]. On the other hand, silica MOFs show lower transmission losses which is required for long distance sensing and moreover, are compatible with telecommunication s equipment. In this current framework, the review is focused on this type MOFs.

In addition to the material used to produce the fibers, there are different manners to classify MOFs for example based on the different geometries, dimensions, distribution and shape of the air holes. One of the most extended criteria considers two different groups: photonic crystal fibers (PCFs) and suspended core fibers (SSCs) (see Figure 1).

PCFs, a term that was proposed in 1995 by Russell et al. [9,32], are characterized by a periodic arrangement of air holes in the section of the fiber. The parameters that define the geometric distribution of the holes are the diameter of the core (ρ), the diameter of the holes (d) and the distance between the centers of two consecutive air cavities (Λ). There are two subcategories depending on the core: hollow core PCFs (HC-PCFs, Figure 2a) or solid core PCFs (SC-PCFs, Figure 2b).

HC-PCFs, which present a negative core cladding refractive index difference (n_core_ < n_cladding_), do not propagate light following TIR mechanism. However, an appropriately designed holes pattern enables light to be guided and confined through a hollow core by the photonic band gap (PBG) principle [33].

Regarding to SC-PCFs (see Figure 2b), the effective refractive index of the cladding will vary with the radial distance, depending on its geometry [34]: actually, the effective cladding refractive index is lower than the one of the core (n_core_ > n_cladding_), enabling TIR mechanism. Since the light guiding properties of SC-PCFs are not a consequence of a material difference (as in SMF) but from an arrangement of a periodic pattern of air holes, the guidance mechanism is known as modified TIR. Although, experimentally, the TIR-guided modes dominate in the majority of SC-PCFs, under the right conditions (high air-filling fraction), TIR and PBG effect could coexist in SC-PCFs [35]; one example is the MOFs called positive core–cladding index difference [36].

The second group of MOFs is SSCs. This term is used to describe MOFs in which relatively large air holes surround a small core (typically a few microns diameter) that seems to be suspended along the fiber length, but it is hold by thin bridges. The light is guided through the core and it is not affected by the periodicity of the holes, so that no band-gap effect is utilized to bound the electromagnetic field. Differently from the PCFs, the guidance properties of SSCs can be explained by TIR theory, as standard fibers, since the condition of n_core_ > n_cladding_ is fulfilled: in this case, n_cladding_ is replaced by n_air_. Figure 3 shows the correct field propagation of the LP_01_ in a specific SSC.

One of the most promising applications for MOFs is the detection of gasses and volatile organic compounds (VOCs). Thanks to the presence of the holes, the molecules of gasses and VOCs can interact directly with the guided light and, in other cases, with the sensing film deposited through the MOFs. The development of applications related to the detection of gasses and VOCs using MOFs is experiencing a considerable growth. In turn, due to the wide variety of different MOF types that exist, the mechanisms of transduction on which the MOF sensors are based are different and depend on the MOF used. For these reasons, firstly, in this review a classification of the different mechanisms of transduction used to develop MOF sensors is presented in Section 2; an overview of the most relevant and recent publications in which the authors used MOFs with the aim of detecting gasses is described in Section 3, whereas Section 4 and Section 5 show a summary about humidity and VOCs sensors, respectively. Finally, the most important ideas derived from this review are presented in the last section.

## 2. Transduction Mechanisms of Gas and VOC MOF Sensors

The interaction between the analyte to be measured and the signal that is registered from the sensor is the transduction. A straightforward classification in two groups consists of considering whether a sensing material is required or not for the interaction [37]. There are different criteria to make subgroups in each category considering the physic or chemical parameter that affects the optical signal, but it is beyond the aim of this review. In this section, a summary about the fundamentals of each interaction is described to have a reference when describing the different sensors from Section 3, Section 4 and Section 5. More details about the transduction mechanism are available in specialized references [38,39,40].

One of the most popular transduction methods used in the development of gas OFS is based on absorption bands. Every gas shows certain absorption bands centered at specific wavelengths so that, a random optical spectrum of a sensor, in presence of a gas or a VOC, will have specific optical losses at the absorption wavelengths associated with it [41,42]: the monitoring of these losses makes possible their identification and quantification. These intensity changes are measured in dB, so that the sensitivity is expressed in dB per ppm or % of the saturated concentration of the gas to be detected. No sensing material is needed and moreover, this mechanism has a high selectivity combined with an enhanced sensitivity as the gas molecules are diffused into the holes of the fiber. The optical signal used to interrogate the sensor has to be as narrow as possible to match the absorption line and not be affected by other spectral components, so that expensive light sources, such as lasers, are required [43,44,45,46,47]. An optical spectrum analyzer (OSA) can be used to register a whole spectral response, although a power meter centered at the absorption line under study can be employed instead: this last option is more affordable, but less versatile.

Another important transduction mechanism is based on the variation in the effective refractive index of the surrounding media of the sensor. The signal guided by the cladding modes interacts with this media and this transduction produces changes in the optical spectrum of the sensor. These variations can be expressed in terms of intensity changes, spectral shifts or spectral shape modification: in these last cases, data processing such as wavelength monitoring or Fast Fourier Transform (FFT) might be required [48]. Measurements based on spectral shifts are more robust because they are not affected by intensity artifacts, but a clear spectral footprint is necessary to define a specific reference (such a narrow band peak) to register the change. Most of the MOF sensors that belong to this group are based on interferometers (Fabry–Pèrot (FP), Sagnac, Mach–Zenhder (MZ), among others) because it is easy to define a reference in the interferometric patterns [34]. In these setups, a sensing material can be attached along the cladding of the fiber or even inside the holes following different deposition techniques. The sensing material determines the sensitivity to gases or VOCs and therefore, the selectivity. In general, to get a spectral pattern it is normally necessary to interrogate the sensor with a broadband source and register the response using a spectrometer [49,50]; however, intensity changes can be also used for the characterization at specific wavelength [51]. There is a wide variety of characterizations depending on signal modulation and they are determined mainly by the set up and the sensing material.

## 3. Gas Sensors Based on MOFs

The development of sensors to detect gases is experiencing great growth due to two reasons [52]: firstly, some gasses are toxic for humans and they are hazardous for the environment; secondly, there are several sectors in the industry that require the detection of different gasses and VOCs to monitor maturing, fermentation processes and other chemical reactions [53,54].

MOFs, more specifically HC-PCFs, were proposed for gas detection in 1999, when it was explained that a significant fraction of the optical power of a holey fiber can be coupled to its own holes allowing, in this manner, the interaction of light with gasses, liquids or sensing films by evanescent field effects [55]. From then on, numerous scientific contributions have proposed the use of MOFs to detect gasses.

### 3.1. MOF Gas Sensors Based on Absorption Bands

The geometry of MOFs easies the diffusion of gasses and liquids and so, the interaction between the optical signal and them. Therefore, in MOFs that use absorption bands, the fiber acts as a “gas cell”. To obtain a proper gas diffusion through the air holes is not an easy process. Several research groups are trying to develop sets-up to make the optimize the diffusion in terms of the geometry or the length of the MOF. For this reason, Y.L. Hoo et al. presented a theoretical and experimental study of acetylene and methane diffusion dynamics based on the pattern of an SC-PCF air holes. Results showed that the final features of the sensor are determined mainly by the size of the holes and the distance between them. After studying all the dimensions chosen by Y. L. Hoo, the optimal length for the MOF sensor developed was set at 4.87 m. With this length, an acetylene sensor was performed with a response time of 1 min and an approximate limit of detection (LOD) of 6 ppm (m/m) [56]. The sensitivity was expressed in parts per million (ppm), parts per billion (ppb) or in % (considering an acetylene saturated atmosphere as 100%).

Acetylene (C_2_H_2_) is one of the most studied gasses because one of its absorption lines is located at 1530 nm, which matches the third telecommunication window: safety regulations set that there is an explosion risk above a 2% concentration. R. Thapa et al. proposed a system based on saturated absorption spectroscopy of the acetylene band near 1532 nm inside two HC-PCFs with core diameters of 10 and 20 μm, respectively. This paper concludes that the variations in the background light transmission, attributed to surface modes, are significantly reduced in the 20 μm fiber due to the set-up used and the structure of the HC-PCF selected. Moreover, the diffusion of acetylene in this structure, fixing the optimal length of the HC-PCF in 0.8 m, improves the interaction with the evanescent field which enhances the sensitivity [57]. Y.L. Hoo et al. reported a study of acetylene diffusion in HC-PCFs [58]: one end of a 10 cm HC-PCF segment was spliced to a SMF and the other one was aligned to an MMF. The set up was placed inside a gas chamber with the aim of coupling as much optical power as possible into the MOF. Due to the space between the two ends, acetylene was able to penetrate the HC-PCF holes. The study concluded that compared with the capillary technique for the diffusion measurement, HC-PCF (based on PBG principle) has a much higher percentage of light power (∼95.45%) that interacts with the sample gas and produces a better sensitivity. Following this idea, in [59], a 10 meters long HC-PCF segment was used as a gas cell filling their holes of acetylene in a photothermal interferometry system, obtaining an LOD of 2 ppb. In [60], a high finesse hollow-core PBG resonating FP gas cell was presented as an MOF sensor to detect acetylene. The gas cell was made with a piece of HC-PCF sandwiched between two SMF segments whose ends were mirrored to achieve high reflectivity. Each of the fibers was inserted in a mechanical ferrule whose function was to optimize the alignment between them and thus ensure the coupling of the light but, in such a way that there was an air hole at both ends of the HC-PCF-SMF alignments. Thus, when the sensor was placed into the gas chamber, acetylene could penetrate and exit of the fiber holes and consequently, the optical power transmitted, associated with the acetylene absorption band, changed. The use of a high finesse resonating HC-PCF cavity reduced significantly the effect of modal interference on gas detection and improved the final sensitivity, obtaining an LOD of 7.5 ppm towards acetylene. Ed Austin et al. also reported the use of an HC-PCF as a gas cell (see Figure 4) [61]. The diffusion of the gas was carried out as it follows: one end of the HC-PCF was spliced to an SMF fiber whereas the free end was placed for several hours in a cell with acetylene, enabling the free diffusion of the gas. After that, the open end was cleaved and spliced to a second SMF pigtail within 5 min: this section was the reference MOF and compared to the absorption lines registered in a gas cell with acetylene concentration varying from 0% to 100% in increments of 10% every 5 min was produced and directed through the measurement gas cell. The sensor showed an LOD of 100 ppm (m/m).

SC-PCFs were also used as gas cells in set-ups to measure acetylene. S.-G. Li et al. demonstrated absorption transmission spectrum of acetylene in a 16.9 cm long index-guiding SC-PCF. The authors mentioned that an important factor to be optimized in order to improve final sensitivity, is to increase the fraction of optical power in PCF cladding air holes. For this reason, they studied it as a function of the index-guiding SC-PCF parameters, inferring that an SC-PCF with small spacing and a large air-filling ratio had a higher fraction of power in its cladding air holes [62]. Based on these results, G. Pickrell et al. developed a process to produce a random-hole distribution SC-PCF using a novel in situ bubble formation technique to increase the interaction areas. The diffusion of acetylene gas was studied in this new structure and the absorption lines of this gas in the near-IR region were detected through the evanescent field of the guided mode in the porous region [63].

Y.L. Hoo et al. reported in [64] an experimental demonstration of evanescent-wave gas detection with a silica-air SC-PCF showing promising results. In order to optimize the optical power that could interact with the measured gasses, the proposed system consisted of the alignment of an SMF with an SC-PCF: the fibers were misaligned and a small part of the SC-PCF was placed into a gas chamber with acetylene so that the gas got into the system. Thereafter, the fibers were re-aligned to carry out the measurements. The system enabled long interaction length making possible the acetylene gas detection: the optical power fraction that could interact with the gas was increased a 6%. This sensitivity, in quantitative terms, was 50 times better than the sensitivity obtained using type D polished fibers and 65 times better than using a SMF. Continuing with the use of SC-PCFs, G. Yan et al. proposed a sensor based on Bragg Grating written on an SC-PCF [65]. The sensor was specially designed to have a photosensitive core and holey cladding for grating fabrication and gas detection. The micro-holes of the SC-PCF served as gas cells, in which the acetylene molecules interact with light through the evanescent field. The Bragg Grating was designed to have a reflected wavelength that matched the acetylene absorption line: the resulting sensitivity was 0.022 dB/%.

A.S. Weeb et al. proposed the use of a SSC to detect acetylene: this structure has a higher air-filling fraction than most of holey fibers, which is relevant for evanescent-field-sensing applications. The system chosen to introduce the gas into the MOF was very similar to those already explained in [58], just changing the use of an HC-PCF for a SSC: the use of this type of MOF showed that the overlap between the different propagated modes was greater than 29% at 1550 nm for a core diameter equal to 0.8 nm, which betters previous coupling ratios [66].

An important conclusion derived from these papers is that the manufacturing of holes along the MOFs and the connection between SMF/MMF and MOF are parameters that must be optimized in each case. Related to the first factor, S.H. Kassani et al. proposed a system formed by a C-type fiber and a Ge-doped ring defect HC-PCF that showed a higher refractive index on the center of the fiber which increased the light coupled [67]. Figure 5 shows the schematic diagram of the proposed acetylene gas sensor device. The C-type fiber segment served as a compact gas inlet/outlet directly spliced to HC-PCF, which overcame previous limitations in sensing capability and dynamic responses due to the rate of gas diffusion into the micron-sized holes. The sensitivity was 4 times higher than the ones of previous set-ups.

Additionally, J.P. Parry et al. addressed this topic in [68]. In this case, the solution proposed to improve the diffusion of the gas was to perform several perforations along the length of the HC-PCF (cone-shaped) using an optical laser to facilitate the gas flow and so, the optical power change at the absorption lines. This set up was able to measure acetylene concentrations lower than 0.05%.

Methane (CH_4_) is also a target gas for sensing applications due to the location of one its absorption spectral lines at the second and third telecommunication windows. A.M. Cubillas et al. demonstrated methane sensing with an HC-PCF in the absorption line centered at 1330 nm. Methane penetrated through the HC-PCF thanks to a small air gap that was left between one end of the PCF and the another one of the SMF, which was angled to avoid Fresnel reflections; LOD equal to 49 ppm (*v*/*v*) was reported [69]. Following the same research line, Y.L. Hoo et al. demonstrated a response optical fiber methane sensor by use of an HC-PCF with periodic microchannels [70]. The authors exposed that the side holes introduced low losses and therefore, it would be possible to construct MOF gas sensors with a sensing length up to a few tens of meters to achieve distributed sensing without compromising the response time. The measurements were performed at methane gas absorption lines close to 1670 nm, achieving an LOD of 647 ppm (m/m). A.M. Cubillas et al. also demonstrated methane sensing with a similar HC-PCF in the same absorption line (1670 nm): a 10 ppm (*v*/*v*) LOD was achieved [71]. The next two scientific contributions presented in this review are a study of two modified HC-PCF structures: in [72], the authors presented a MOF gas sensor based on an octagonal geometric structured HC-PCF. They studied the optical properties of the proposed fiber numerically by full vector finite element method (FEM) using the software COMSOL. The octagonal lattice of the HC-PCF was optimized to obtain better relative sensitivity and low confinement loss acquiring the maximum relative sensitivity of around 97% and minimum confinement loss of ~0.007 dB/m at 1670 nm wavelength where the absorption line is located; on the other hand, in [73], a gas sensor based on spiral HC-PCF (see Figure 6) was presented. Again, in order to optimize its structure, a numerical analysis was performed by utilizing FEM, varying the geometrical parameters to obtain an optimized sensitivity at the frequencies close to the absorption lines of methane within the range 1000–1800 nm. To conclude with MOF methane sensors, L. Kornaszewski et al. detailed in [74] a sensor using an HC-PCF based gas cell and a broadband source. The holes of the HC-PCF were filled with the gas and Fourier Transform spectroscopy was used to measure transmission spectra in the 3.15–3.35 μm methane absorption region, achieving to measure concentrations below 0.1%.

Hydrogen (H_2_) is also an important gas to detect and there some studies in the bibliography about its monitoring with MOFs. F. Yang and W. Jin reported in [75] a highly sensitive all-fiber hydrogen sensor based on continuous-wave stimulated Raman gain spectroscopy with an HC-PCF operating around 1550 nm. The output end of the sensing HC-PCF was spliced to a SMF preventing the collapse of air-holes while the input end was coupled to an input SMF with a mechanical splicer. Thanks to the light scattered by the H_2_ molecules, at certain frequencies there was an increase in the optical power which depended on the concentration of H_2_ molecules. In this paper, the HC-PCF worked again as a cell and the gas detected was pressurized into the HC-PCF through the gap between HC-PCF and an input SMF. The experiments demonstrated an LOD of 17 ppm (m/m) of hydrogen; due to the transmission window of the fiber used, the proposed system could be used to detect other gasses such us N_2_, O_2_ and CO_2_. The same set-up and MOF sensor were used by Riccardo Pennetta et al. but in this case the splice between the SMF and the end of the HC-PCF was not directly exposed to the gasses: it was protected with a thermally fixed cylinder formed by a silica nanospike. Thanks to this alternative, the final device was much more compact and stable compared to free-space arrangements allowing an efficient coupling with only 1–2 dB insertion losses [76]. V.P. Minkovich et al. proposed a hydrogen MOF sensor that was based on the use of a tapered SC-PCF with collapsed air-holes coated with a thin palladium layer, improving the final sensitivity (see Figure 7). The collapsing of the holes increased the evanescent field affected by the gas-permeable thin film which produced absorption changes. The resulting optical power transmitted was increased by a factor of 4 offering a wider absorption lines dynamic range [77].

Nitrogen (N_2_) is another gas that has been detected using MOF. M.P. Buric et al. proposed the use of spontaneous Raman backscattering to detect low pressure molecular nitrogen in the range of 100 ppm (m/m). The authors described that an improvement on the sensor sensitivity was possible thanks to the use of an HC-PCF in the backscattering configuration increasing the coupling efficiency and reducing silica-Raman background noise. The HC-PCF worked as a gas cell: this set-up enhanced the interaction area between the light guided and nitrogen molecules, which improved the final sensitivity [78].

Oxygen (O_2_) is another gas that researchers have detected using MOFs. X. Yang et al. in [79] proposed an oxygen gas optrode by forming a fluorophore doped sensing film. Organic silicate gels and plasticized cellulose acetate were chosen as sensing film and they were deposited inside the microholes of an SC-PCF. In this paper, the sensing material emitted fluorescence into the PCF segment and that was the signal used to characterize the sensor. Ratios between the luminescent intensity with a 0% and 100% concentration equal to 1.51, 2.81 and 10.8 were reported for the different sensing films.

The detection of carbon monoxide (CO) using MOFs is reported in [80] by H. Ding et al. The detection of this gas is quite relevant because of its toxicity and dangerousness for humans. An all-fiber compact gas sensing system using an HC-PCF as a gas cell was developed showing relatively low transmission losses. The proposed set-up consisted of an HC-PCF, at whose ends, an SMF and an MMF were aligned, leaving in both interfaces an air gap between them; the gas was inserted through the air gap formed between the SMF and the HC-PCF, while the other end was used to empty the system. The properties of the proposed system were demonstrated experimentally by detection of CO and acetylene registering approximately linear relationships: both gasses could be measured using a tunable laser used that matched their absorption bands. The LODs were 300 ppm (m/m) and 5 ppm (m/m) for CO and acetylene, respectively.

Other works report the detection of at least two gases with the same system: as an example, hydrogen cyanide (H^13^CN) and acetylene were measured using a single sensor. The detection of HCN is important because it is acts as a precursor for amino and nucleic acids [81]. J. Henningsen et al. proposed the study of the saturated absorption in overtone transitions of theses gas molecules confined in an HC-PCF. MOF was used as gas cell to monitor the power located at their absorption bands (close to 1530 nm) [82]. Following this idea, T. Ritari et al. proposed the detection of several gasses: acetylene, hydrogen cyanide, methane and ammonia (NH_3_) could be detected using a tunable gas that matched their absorption lines [83]. The inlet and outlet of the sensor used the air bandgaps as it was described before.

B.K. Paul et al. proposed in [84] a new MOF structure that improves the properties and features of previous gas sensors, in terms of nonlinearity, confinement loss, sensitivity and splice loss of the modeled MOF (see Figure 8). To achieve this, a rigorous numerical analysis was done in spectral range from 1.30 μm to 2.0 μm. The study concluded that a good solution was to make several slots on the core of the SC-PCF: the results were checked out by exposing the sensor to diazene (H_2_N_2_). Y. Hao et al. proposed a novel design of an antiresonant (AR) nodeless HC-PCF, which could enable gas samples to diffuse in or out of the fiber optic through the open cladding in a very short time. The unique properties of the MOF used such as mode pattern, fraction of power in the central core, confinement loss and response time of the fiber were investigated in detail. The paper inferred that this design had extremely high overlap between light and gas samples and wide low-loss transmission windows in visible and near-infrared wavelength bands. Furthermore, the gas samples could diffuse in and out of the air core without destroying the waveguide structure, and the open channel will not arise excess loss. These advantages will make it very useful for advanced gas sensing with high sensitivity and quick response [85].

J.K. Ranka et al. analyzed in [86] the waveguide properties of an SC-PCF consisting of a silica core surrounded by a single ring of large air holes (see Figure 9). Although these fibers can support numerous transverse spatial modes, coupling between these modes even in the presence of large perturbations is prevented for small core dimensions, owing to a large wave–vector mismatch between the lowest-order modes. The result is an SC-PCF that couples single-mode with propagation properties that can be achieved only in multi-mode waveguides. In [87] the authors proposed to develop a microstructured core inside of an SC-PCF core for gas sensing applications. The core contained vertically arranged elliptical holes in order to enhance both sensitivity and birefringence. Structural geometric parameters were also optimized.

To conclude this section, in [88], the usability, advantages and limitations of SC and HC-PCF for gas sensing in the NIR were discussed. SC-PCF of various geometries and HC-PCF with different transmission properties were evaluated considering their sensitivity towards combustion gasses such as methane and acetylene. Although both types of fiber can be used in gas detection applications, the transduction mechanism of HC-PCFs make them relevant for detecting very low gas concentrations, in which short fiber lengths (a few meters) are used. On the other hand, the transduction mechanism of SSCs makes their use more advantageous for detecting high gas concentrations and sensors with longer lengths, which is useful for distributed sensing.

### 3.2. MOF Gas Sensors Based on of Effective Refractive Index Variations in the Surrounding Medium 

Gasses have the property of altering the effective refractive index depending on their chemical properties and concentration. In the next paragraphs, the most relevant papers found in the bibliography are described.

H. Liu et al. proposed the simultaneous detection of hydrogen and methane when they were mixed, thanks to a special structure based on an SC-PCF: this MOF consisted of four ultra-large side-holes distributed vertically and horizontally together to an array of smaller air-holes along the angle of 45° and 135° of the cross-section (see Figure 10a). A gold thin film was deposited following sputtering technique inside the holes with the aim to generate a surface plasmon resonance (SPR). Over this layer, another two different thin films were also deposited: Pd-WO_3_ (in hole with number 1), which was sensitive to hydrogen and fluoro-siloxane (in the hole with number 2), deposited by means of dip coating technique, which was reactive to methane (see Figure 10b). In the presence of the gases, a wavelength shift at the SPR frequency was observed. The proposed sensor had a sensitivity of −1.99 nm/% for methane and −0.19 nm/% for hydrogen without any interferences [89].

In [90], a MZ interferometer based on a slotted SC-PCF was developed for measurements of refractive index variations. The sensor consisted of a section of six-air-hole grapefruit PCF sandwiched between two SMFs; after splicing them, the air holes were collapsed generating a MZ interferometer. Then, one of the PCF holes was opened via two micro slots made by a femtosecond laser micromachining without compromising the optical waveguide robustness. The sensor was exposed to nitrogen and when the gas molecules filled the open air-hole, a large overlapping of the mode field and gas sample produced a sensitivity of −827.94 dB/RIU. The same idea was used by A.M. Shrivastav et al. in [91] to develop a MZ interferometer but the SC-PCF was formed by a periodic pattern of air holes. A tin dioxide (SnO_2_) sensitive film was firstly deposited in the holes and thereafter, a polyaniline (PANI) thin layer. SnO_2_ has shown a great potential for the development of chemical and biological sensors due to redox reactions with various analytes; on the other hand, PANI is a promising candidate for gaseous sensing applications. The sensor had an LOD of 8.09 parts per trillion (ppt) for ammonia.

Table 1 summarizes the most relevant features (sensitivity and LOD) of the MOF gas sensors listed in this section.

## 4. Humidity Sensors Based on MOFs

Relative humidity (RH) is an important parameter that is required in several applications. RH is defined as the ratio between the current water vapor pressure and the saturated one at a certain temperature [92]. Taking this definition into account, humidity could be considered as a vapor, so that the reported MOF sensors to measure it are considered in this review: they are organized considering the transduction mechanism.

### 4.1. MOF RH Sensors Based on Humidity Absorption Lines

As in the case of gasses of the previous section, HC-PCF were also used as “humidity chamber” basing its transduction mechanism on the absorption band of water vapor center at 1368.59 nm. M.Y.M. Noor et al. proposed a novel MOF RH sensor based on the air guided by an HC-PCF using the direct absorption spectroscopic method [93,94]. The humidity level was determined by the optical power difference between a reference signal and a MOF sensor signal centered at the peak of water vapor absorption. The authors demonstrated that setting a small air gap between SMF and HC-PCF yield to a resolution equal to 0.2%RH in the range from 0 to 90%RH with no sensing material.

### 4.2. RH Sensors Based on Effective Refractive Index Changes of the Surrounding Medium

In this first part, sensors based on SC-PCF and MZ interferometers in transmission configuration will be exposed. The interferometer is developed by means of splicing an SC-PCF segment between two SMFs pigtails. The splicing is carried out in such a way that the voids of the SC-PCF are collapsed completely along a short region (see Figure 11). At this point, part of the fundamental SMF mode is coupled to the SC-PCF cladding modes which are affected by the external refractive index changes. The propagation constants of PCF cladding modes are different and, consequently, they accumulate a phase difference as they propagate along the SC-PCF section. Once these cladding modes reach the second PCF-SMF transition, they interfere with the remaining fundamental SMF mode, producing an interferometric response. Based on this structure, M. Y. Mohd Noor et al. proposed an SC-PCF interferometer RH sensor in [95]: a schematic drawing is shown in Figure 12. Thanks to the effect of the collapsed region described above, a certain part of the evanescent field of the light guided could interact with water molecules, which altered the effective refractive index of the surrounding medium. Measuring the wavelength shift of one of the pattern peaks generated by the MZ interferometer, a humidity sensitivity of 60.3 pm/%RH in the range of 60–80%RH and 188.3 pm/%RH in the region from 80% to 95%RH was achieved without using any hygroscopic thin film. In addition, the sensor showed a low temperature cross sensitivity (0.5 pm/°C).

To improve the sensitivity, several authors proposed to deposit a hygroscopic thin film using different techniques such as dip coating, layer-by-layer nanoassembly (LbL) or sputtering. Firstly, Tao Li et al. proposed to coat the section of the SC-PCF with a layer of polyvinyl alcohol (PVA) [96]. Experimental results showed that a wavelength blue shift of 40.9 pm/%RH was achieved within a measurement range of 20–95%RH. In the same manner, P. Wang et al. proposed to use a film of methylcellulose with a MZ interferometric pattern wavelength red shift of 10 nm for a RH from 30% to 85%RH [97]. With the same set-up, J. Mathew et al. proposed to deposit an agarose film along the SC-PCF using dip coating technique [98]. The sensor showed a total wavelength blue shift of approximately 56 nm for a humidity range from 40% to 95%RH. The authors divided in two parts the RH study due to the different sensitivity of the MOF sensor: in the first part, between 40% and 80%RH, the MOF sensor had a linear behavior with a sensitivity of 0.57 nm/%RH and a resolution of 0.017%RH; for the second one, in the range of 80–95%RH, the response was exponential and resolution was more accurate (0.007%RH).

The following publications are still based on MZ interferometers using SC-PCF but include some set-up modifications that are remarkable. P. Zhang et al. proposed an MZ interferometer for RH and temperature measurement [99]. The set-up presented is similar to previous ones [90] but a MMF segment is spliced between the SMF and the end of the SC-PCF (see Figure 13). In this manner, it was possible to excite the higher order modes and facilitate the coupling of them to the air holes, improving the final sensitivity of the MOF sensor. The sensitivity to moisture was −0.077 dB/%RH, in a range from 25% to 80%RH and regarding to temperature, the sensor showed a temperature sensitivity of ∼3.3 pm/°C in the range of 25 °C–70 °C.

S. Zhang et al. also presented in [100] simultaneous measurement of RH and temperature: the sensor was formed by the cascade structure SMF-SC-PCF-SMF and a fiber bragg gratings (FBGs) (see Figure 14). The PCF-MZ interferometer and FBGs had different responses to humidity and temperature (blue shift) so that simultaneous measurements were achieved with resolutions of 0.13%RH (humidity range: 30–95%RH) and 1.0 °C (temperature range: 20 °C–70 °C).

R. Gao et al. described a MOF sensor constructed by filling with agarose gel the gap between an aligned SMF and an SC-PCF in [101]. An MZ interferometer was fabricated by splicing the other end of the SC-PCF and another SMF (see Figure 15a). Due to the tunable refractive index property of the agarose gel, the mode field diameter of the propagation light changed with the external relative humidity, which induced the change of coupling ratio between the SC-PCF and the SMF (see Figure 15b): experimental results showed a sensitivity up to 2.2 dB/%RH between 20% and 80%RH.

R. Tong et al. proposed in [102] a new MZ interferometer structure including a tapered fiber before and after collapsed regions (the two stretched segments acted as a mode splitter/mixer). This configuration was coated with graphene quantum dots (GQDs) and PVA. Sensing properties of different thicknesses of GQDs-PVA films (2.48, 3.72, 4.45, 5.96, 7.42 and 8.17 µm) were investigated experimentally, and the corresponding sensitivities obtained were 0.0901, −0.0797, −0.0337, 0.0586, 0.0539 and 0.0313 nm/%RH, respectively when the measurement range changed from 13.77%RH to 77.87%RH. These results revealed that when a sensitive thin film was deposited along the MOF sensor, its thickness is critical for the final sensitivity, selectivity and response/recovery times. Taking it into account, J. Mathew et al. reported in [103] a study of the effect of the agarose coating thicknesses on the RH sensor sensitivity based on an SC-PCF MZ interferometer. According to the comment exposed above, this paper also inferred that the RH sensitivity of the MOF sensor had a significant dependence on the thickness of the agarose coating (see Table 2). Two similar studies were developed by Diego Lopez-Torres et al. but in this case, the nanofilms were made, in the first work, with a polymer nanolayer formed by poly(allylamine hydrochloride) (PAH) and poly(acrylic acid) (PAA) [104] and, in the second one, by SnO_2_ [105]. In [104], the optimal MOF sensor, with a polymer nanolayer of 240 ± 20 nm, showed a humidity resolution of 0.074% and operated in the 20–95%RH range. Regarding to [105], the optimal SnO_2_ thickness for the MOF sensor was 1150 nm showing a wavelength red shift of 67 nm with a sensitivity of 0.96 nm/HR% and a humidity resolution of 0.067%HR for RH values between 20% and 90%.

SC-PCFs working in reflection mode were also used in several papers for RH sensing. This configuration consisted of making the same splice between SMF and SC-PCF and leaving the other end of this last fiber open. When the cladding and core modes reach the opened interface, they are reflected back to collapsed region and, at this point, they were recombined as a single mode into the core of the SMF. Based on this configuration, a Michelson interferometer is generated. S.S. Hindal and H.J Taher presented a sensor based on this configuration with no sensing material [106]. The final sensitivity obtained was 2.41 pm/%RH for RH range between 27% and 85%. Only varying the length of the SC-PCF used and reducing the losses introduced by the splices, J. Mathew et al. obtained a final sensitivity of the sensor [107]. In the humidity range between 40–70%RH, the measured sensitivity was 6.6 pm/%RH, while for values above 70%RH, the sensitivity was 24 pm/%RH.

A hygroscopic polymer, such as agarose, can be deposited in the microholes of the free end in other to improve the final sensitivity of the MOF sensor. In [108], this idea was proposed, characterizing the sensor response in terms of reflected optical power variations instead of the wavelength shifts. For a humidity range from 14% to 98%RH, changes around 12 dB were registered. Continuing with this idea, Jinesh Mathew et al. described in [109] an optical fiber hybrid device for simultaneously measuring temperature and RH. The device was formed by a FBGs and a reflection SC-PCF Michelson interferometer infiltrated with agarose: the FBGs were used to monitor the temperature whereas the sensing film was sensitive to humidity. The resulting device showed a temperature sensitivity of 9.8 pm/°C and an optical power variation of ~7 dB for an RH change from 20% to 95%RH.

A highly reproducible and cost effective sensing platform that could be useful for sensing diverse parameters was proposed in [110]. The MOF sensor was made by splicing both ends of the SC-PCF to two SMF segments but in this case, it was based on in reflection mode because the second segment spliced was a short SMF region cleaved and coated to develop a mirror. The length and section geometry of the SC-PCF used defined the interferometric pattern, so that it could be used for RH sensing. Taking this idea into account, a simple sealed-void PCF interferometer humidity sensor was reported in [111] by M.Y.M. Noor et al. Due to the adsorption and desorption of H_2_O molecules, a wavelength red shift occurred in the reflection spectrum of the sensor. The sensor exhibited a sensitivity between 20.3 and 61.6 pm/%RH in the ranges of 60%RH–80%RH and 80%RH–95%RH, respectively. W. Wong et al. proposed in [112,113] the same reflection set up but depositing PVA because of its hygroscopic nature to detect RH variations (see Figure 16): this sensor showed a sensitivity of 0.60 nm/%RH, from 30%RH to 90%RH. F.C. Favero et al. proposed in [114] a device based on this configuration used to control the breathing of a person. It could be particularly important in applications where electronic sensors fail or are not recommended. It may also be useful in the evaluation of a person health and even in the diagnosis and study of the progression of serious illnesses such as sleep apnea syndrome.

S. Zheng et al. report the inscription of a long-period gratings (LPGs) in an SC-PCF to get a reflection set-up [115,116]. The holes of the SC-PCF-LPGs were coated with two types of nanostructured polymer films: PAH^+^/PAA^−^ and alumina (Al_2_O_3_^+^)/poly(sodium-p-styrenesulfonate) (PSS^-^), respectively by LbL deposition technique. The first coating was used to increase the sensitivity to refractive index variations of H_2_O in the air channels (swelling/deswelling) whereas the Al_2_O_3_/PSS coating was used for the selective absorption of H_2_O. The sensor showed a wavelength shift of 0.058 nm/%RH and an optical power variation of 2 dB for a humidity range from 20% to 54%RH.

There are also papers that report using HC-PCFs to develop RH sensors in the bibliography. The most used configuration is based on a reflection FP interferometer. The response is obtained by splicing a section of an HC-PCF with a SMF segment without collapsing its holes and then, depositing a hygroscopic material on the other HC-PCF end. In this manner, due to the differences in the refractive indices, 3 interfaces were generated: SMF/HC-PCF, HC-PCF/hygroscopic polymer and hygroscopic polymer/air. The reflected rays at these points interfered between them and determined the interferometric response. Based on this explanation, a RH MOF based on FP interferometry configuration was presented by L.H. Chen in [117] (see Figure 17). The proposed sensor was functionalized with a thin layer of a moisture-sensitive natural polymer chitosan to form a low fineness FP sensor. The MOF sensor showed a sensitivity of 0.13 nm/%RH for RH ranging from 20%RH to 95%RH with a RH uncertainty of ±1.68%RH and fast response time of 380 ms. H. Sun et al. also proposed and demonstrated a simultaneous RH and temperature MOF sensor [118]: this device was constructed by splicing a short length of HC-PCF to SMF, but now, a thin coating of PVA was deposited onto the HC-PCF cleaved surface. Experimental results demonstrated that this sensor could simultaneously measure the ambient RH and temperature. The sensor showed a maximum wavelength red shift of 5 nm for RH variations between 35–95%RH and 0.8 nm for temperature changes from 20 °C to 80 °C. X. Liu et al. proposed a similar set up but using chitosan as sensing material with the aim of improving the final sensitivity [119]. The FP interferometer sensor showed a response with a maximum sensitivity of 0.28 nm/%RH at high RH level, close to 90%RH.

Q. Sui et al. proposed in [120] a RH sensor based on two cascaded FP interferometers. It was constructed by splicing a segment of HC-PCF between two sections of SMF. By means of the LbL technique, a polymer multilayer of PSS/PAH was deposited on the end face of the FP interferometer. Due to the variation of the RH, the effective refractive index of the thin film changed as well as its thickness, and consequently, the reflected optical power. This difference was measured, and the MOF sensor showed a final sensitivity of 0.008 dB/%RH in a RH range between 5–90%RH. In order to improve this value, Y. Zhao introduced some changes in this configuration: the FP interferometer was constructed by splicing a SMF and SC-PCF segments as it is displayed in Figure 18 [121]. Moreover, the HC-PCF was partially filled with GQDs and PVA. Here, the sensitivity of the MOF sensor obtained was 0.456 nm/%RH with RH with a dynamic range from 19.63 to 78.86%RH.

Theoretical simulations based on an HC-PCF RH sensor were presented in [122] to verify its behavior for humidity sensing. In this configuration, the MOF was used as the sensing element which was tapered and filled with a moisture sensitive polymer (hydrogel). The fiber consisted of a solid germanium doped core which was surrounded by five large air holes. The core mode in the tapered HC-PCF was very sensitive to any index change at the air-holes cladding interface because of the large modal field interaction with the surrounding air holes in the waist of the fiber. The sensing principle was based on the change in the refraction index (RI) of the material when it was exposed to humidity. The results showed that the loss of the filled HC-PCF varied from 0.063 dB/cm to 75.847 dB/cm when the RH ranged from 0 to 95%RH.

This section finishes with RH sensors based on SSCs: a FP interferometer was fabricated by splicing a SMF segment to an SSC whose end was perpendicularly cleaved: A. Lopez-Aldaba et al. proposed in [123] its use. The sensor was based on the deposition of SnO_2_ on a SSC low-finesse FP sensing head by sputtering (see Figure 19). The sensor interrogation was carried out by monitoring the FFT phase variations of the interferometric pattern. The device showed a linear behavior from 20% to 90%RH in which the sensitivity was 0.14 rad/% and the humidity resolution was 0.0026%RH. This sensor was also used by the same authors for real soil moisture measurements in [124]: its performance was compared with a commercial capacitive soil moisture sensor in two different soil scenarios during two weeks. The authors inferred that the MOF sensor showed a great agreement with capacitive sensor response and gravimetric measurements, as well as a fast and reversible response making it very interesting for this type of measurement in real time.

C. Wang et al. also used a SSC to develop a RH sensor [125]. The hybrid cavity of the MOF sensor was consisted of two parts: the first one was a small segment of an SSC formed by large holes around its suspended core; the second one was a hygroscopic optical adhesive (OA) section (see Figure 20). The total length of the hybrid cavity was 240 µm (198 µm length for the FP cavity and 42 µm the thickness for the adhesive). In this way, the response was obtained by the interference between the signals reflected at the three interfaces: SMF/SSC, SSC/OA and OA/air. Due to the different sensitivities at each FP cavity to RH and temperature, both parameters were measured simultaneously by monitoring the phase shifts obtained by the FFT. In the range between 15%-90%RH, the sensor showed a sensitivity of −0.042 rad/%RH; for temperature measurement, the SSC sensor got a sensitivity of −0.052 rad/°C in the range from 15 °C to 40 °C.

Table 3 summarizes the most relevant features (sensitivity, spectral resolution and HR range) of the MOF humidity sensors listed in this section.

## 5. VOC Sensors Based on MOFs

Detection of VOCs plays an important role in daily life because some of them are a significant threat to human health as well as to ecological balance. There is a large number of potential applications based on VOCs detection such as environmental monitoring, chemical and food industry just to list a few [37]. A special mention deserves the applications related to biomedicine which are becoming a topic with a great interest. One of them is the diagnosis of diseases measuring certain indicators, such as VOC molecules, in the exhaled breath. To perform them, and based on [126], in which an overview of the recent developments in VOC detection with micro- and nano-engineered optical fibers is detailed, MOFs are presented as a good alternative due to the possibility of making the sensors more sensitive and selective to different concentrations of distinct VOCs. Moreover, MOFs are especially useful for sensing VOCs because of their dimensions and design which permit diffusion of VOC molecules through it, acting as a cell for the vapors.

One of the main differences in comparison with the detection of gasses is that the absorption bands of VOCs are not located in range of the telecommunication windows. For this reason, the most used transduction mechanism of MOF VOC sensors is based on detecting changes in the effective refractive index of the surrounding medium, although there are some specific set ups based on VOCs absorption bands: an example found in the bibliography is a work by J. M. Kriesel et al.: it is based on a mid-infrared spectroscopy system to measure the concentration of different VOCs with an ultra-low sample volume in the order of pL [127]. This system combined a quantum cascade laser with an HC-PCF gas cell. The laser had enough spectral resolution to measure gasses with narrow absorption bands (e.g., water, methane, ammonia, etc.) as well as VOCs (e.g., aldehydes, ketones, hydrocarbons) whose bands were located at the mid-infrared range is between 2.5 and 25 μm. Thanks to the HC-PCF holes, the beam light emitted by the laser was able to interact with the different molecules of the measured VOCs. Taking this system as a starting point, C. Charlton et al. developed a MOF sensor to detect ethyl chloride (C_2_H_5_Cl) at concentration levels of 30 ppb (*v*/*v*) [128].

Now focusing on the transduction mechanism based on refractive index variations, J. Villatoro et al. reported in [129,130] an in-reflection SC-PCF interferometer which exhibits high sensitivity to different VOCs (acetonitrile (CH_3_CN) and tetrahydrofuran (THF)), without the need of any permeable material (see Figure 21a). The interferometer consisted of an SC-PCF segment spliced to a SMF. In the splice, the voids of the SC-PCF were fully collapsed, thus allowing the excitation and recombination of two core modes (see Figure 21b). The device reflection spectrum exhibited an interferometric pattern that was red shifted when the voids of the SC-PCF were infiltrated with distinct VOC molecules. Experimental tests were carried out for CH_3_CN and THF (by injecting a volume of 35 µL into the measuring cell where the VOCs were evaporated and the sensor was introduced) obtaining a maximum shift of 150 and 820 pm, respectively.

Using also an HC-PCF in reflection mode, B. Kim et al. developed the following sensor configuration [131]: one end of the HC-PCF was spliced to an SMF, while the other end was spliced to an SC-PCF. In this way, a FP interferometer was generated due to the 3 interfaces with different refractive indices that existed. The interference spectrum resulting from the reflected light at the silica and air interfaces changed (in terms of wavelength shifts) when the micro-cavity was filled with acetone vapors. This structure enabled the direct detection of VOCs molecules without the deposition of any sensing material but, due to this fact, it is not possible to distinguish one specific VOC from the rest. To analyze the final sensitivity of the MOF sensor towards acetone (C_3_H_6_O), the FFT technique was used showing a shift of 1.3 rad.

HC-PCFs were also proposed to develop MOF sensors able to detect VOCs but using the transmission configuration. L. Niu et al. proposed an OFS for ethanol detection, based on Rayleigh scattering effect in an HC-PCF [132]. Both ends of HC-PCF were aligned with SMFs without splicing, leaving a very short air gap to enable molecules diffusing. As the ethanol (C_2_H_5_OH) concentration in the gas chamber increased, ethanol molecules came into the air holes of the HC-PCF. In this manner, Rayleigh scattering was excited, and it produced an intensity loss of transmitted light. When the light reached the second SMF, the modes recombined into the fundamental one, working similarly to an MZ interferometer. Experimental results showed a sensor sensitivity to ethanol of 0.022 dB/ppm (m/m). The same authors measured the wavelength shift in the reflected signal spectrum to detect the ethanol again [133]. The registered concentration range was from 250 ppm(m/m) to 1000 ppm (m/m) of ethanol and the wavelength red shift was 2.56 nm.

Regarding to SSCs, there are papers in which these optical fibers have been also used to detect VOCs. A. Lopez-Aldaba et al. multiplexed six FP sensing heads [134]. Each one of them was prepared splicing different length segments of SSC to SMF in order to obtain specific interferometric patters for each device. In three of them FP sensing heads, an indium tin oxide (ITO) thin film was deposited inside the holes of the SSCs to make them more sensitive to ethanol whereas other two were functionalized with SnO_2_ to show a different response; the last one was left with no sensing coating to measure temperature. The interrogation of the sensing heads was carried out by monitoring the FFT phase variations in the interferometric pattern of each sensor. This method enabled to multiplex several MOF sensors because each one showed different interference patterns. A maximum phase shift of 0.31π rad was obtained for saturated atmospheres of methanol (CH_3_OH). With the goal of developing SSC sensors with an improved sensitivity, D. Lopez-Torres et al. studied and compared the sensitivity of FP sensors prepared with SSCs that had cores with different geometries [135] (see Figure 22). The probes were evaluated when exposed to saturated atmospheres of ethanol. This paper concluded that the geometry cavities, but mainly the one of the cores, determined strongly the final sensitivity: an SSC with a hole in the center of its core showed a sensitivity 5 times higher compared to the others.

Table 4 summarizes the most relevant features (sensitivity and LOD) of the MOF VOC sensors listed in this section.

## 6. Conclusions

As it is seen throughout this review, there are many set-ups and configurations in which different types of MOFs have been used to develop VOC, RH or gas sensors. MOFs present some advantages when compared to SMFs due to their structure: they are versatile and many different configurations can be achieved to develop interferometers such as MZ, FP or Michelson with suitable sensor properties such as good kinetics, small size and good time stability. The holes of the MOFs play an important role because they can host gas and VOC molecules improving in this manner the kinetics; moreover, the interaction between guided light and analyte or deposited sensing materials (thin films) is improved, and consequently, the sensitivity and the resolution of the devices.

The two main transduction mechanisms of MOFs are the spectral absorption bands of the gas to be detected and the variation of the effective refractive index of the surrounding medium of the MOF sensor due to the presence of gas or VOC molecules. In the last case, variations in the transmitted or reflected signals can be monitored in terms of optical power (transmitted/reflected) or wavelength/phase shifts.

The majority of the proposed MOF sensors described are an alternative to develop gas or VOC sensors if they are compared to electronic sensors. Table 5 summarizes the analyzed references depending on the type of MOF, transduction mechanism and target magnitude. These publications reinforce the idea that MOF sensors have a relevant potential in the field of OFS: the number of gasses and VOCs that can be detected obtaining optimal results is significant, and moreover, different MOF configurations can be used looking for optimizing the final sensor features. This constitutes a framework that encourages researches to keep on developing gas and VOCs sensors based on MOFs.

## Figures and Tables

**Figure 1 sensors-20-02555-f001:**
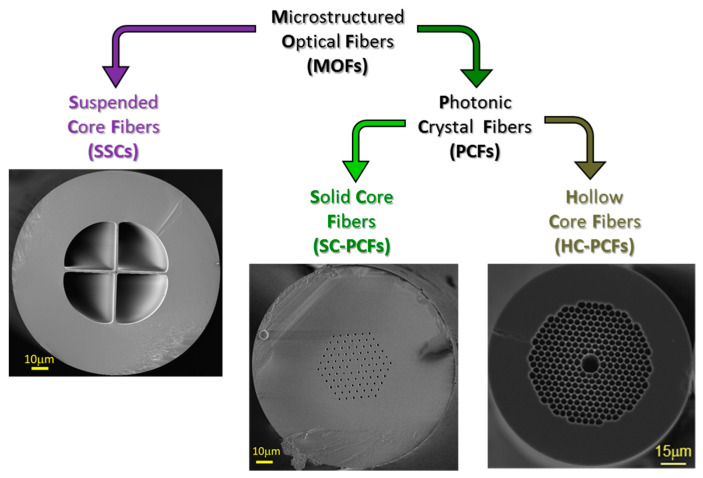
Classification of the different types of microstructured optical fibers (MOFs) based on their structures.

**Figure 2 sensors-20-02555-f002:**
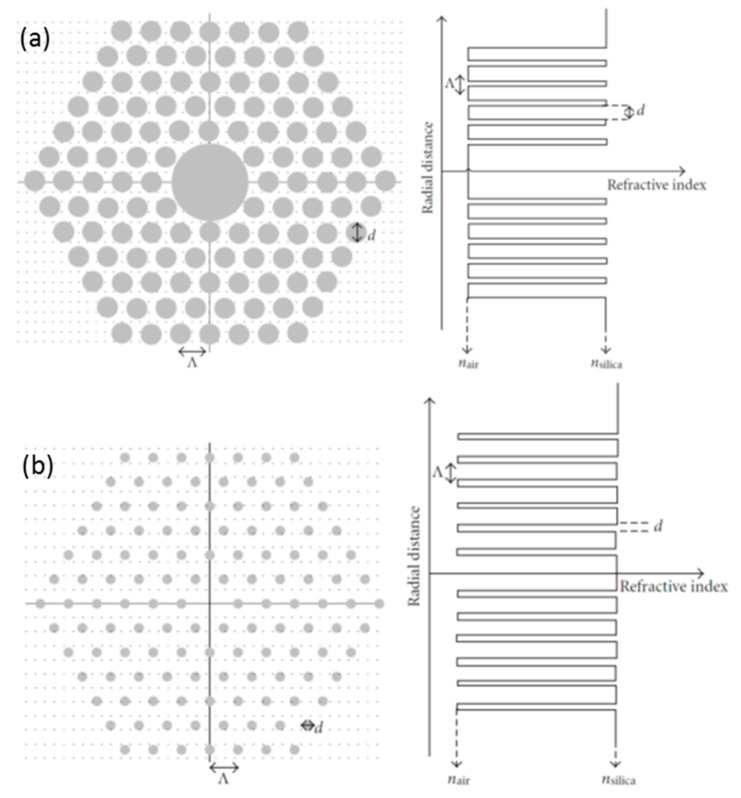
(**a**) Illustration of hollow core (HC)-photonic crystal fibers (PCF) cross-section and respective refractive index profile; (**b**) an illustration of solid core (SC)-PCF cross-section and its refractive index profile. The colors are grey for air and white for silica. Reprinted with permission from [34].

**Figure 3 sensors-20-02555-f003:**
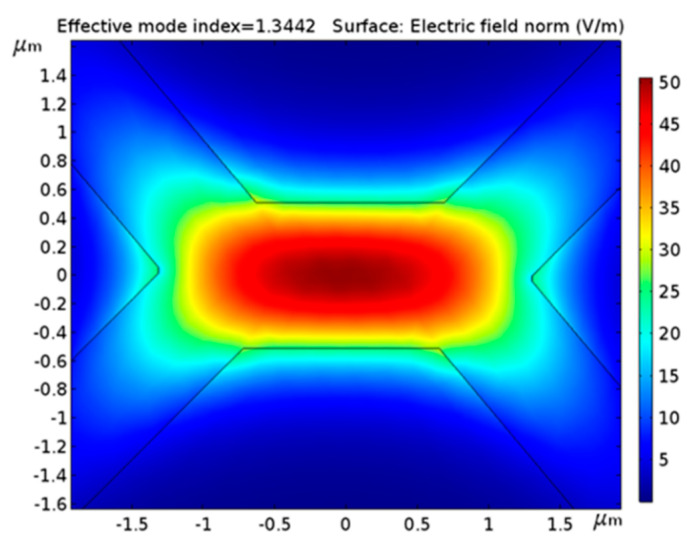
Suspended core fiber (SSC) cross-section LP_01_ mode field distribution based on total internal reflection (TIR) theory.

**Figure 4 sensors-20-02555-f004:**
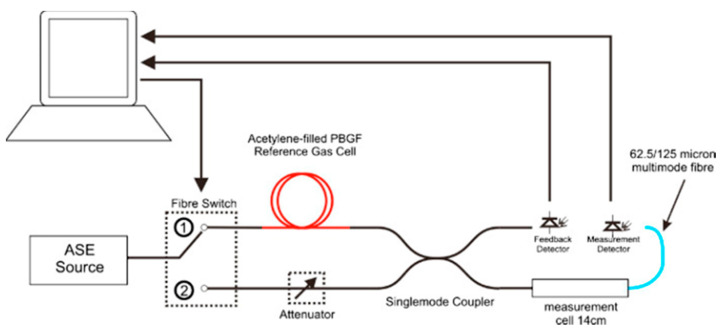
Experimental setup. DFB laser: distributed feedback laser; HC-PCF: hollow core photonic crystal fiber (in the paper figure as called PBGF); SMF: single-mode fiber; MMF: multi-mode fiber; PD: photodetector; PC: personal computer. Reprinted with permission from [61].

**Figure 5 sensors-20-02555-f005:**
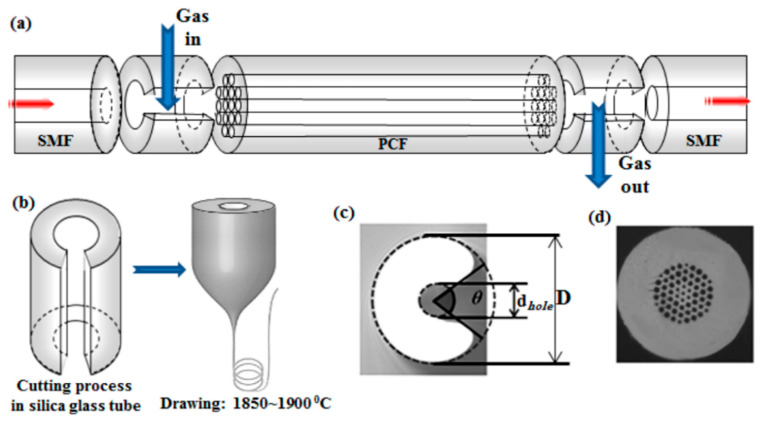
(**a**) Schematic diagram of the proposed gas sensor device including SMF connected to the light source and the detector, PCF as a sensing medium and C-type fibers as inlet/outlet components; (**b**) fabrication process of the C-type fiber; (**c**,**d**) scheme and cross section of the C-type fiber. Reprinted with permission from [67] ©2013 Optical Society of America.

**Figure 6 sensors-20-02555-f006:**
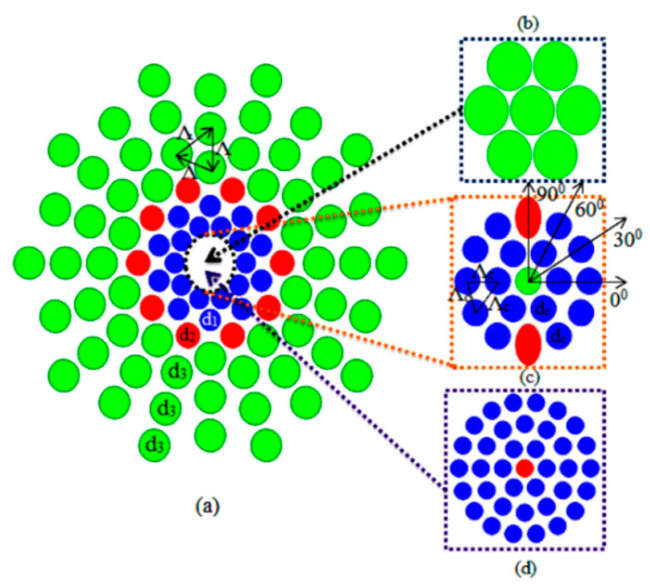
(**a**) Overall structural view of the proposed spiral HC-PCF. The cladding is formed with spiral shape that contains circular air holes; (**b**) air hole details of the first spiral ring of the core S-PCF; (**c**) details of the circular structure of the cluster in the core region which are formed into a microstructure shape; (**d**) core region magnification of the enlarged end face view of the suggested S-PCF. Reprinted with permission from [73].

**Figure 7 sensors-20-02555-f007:**
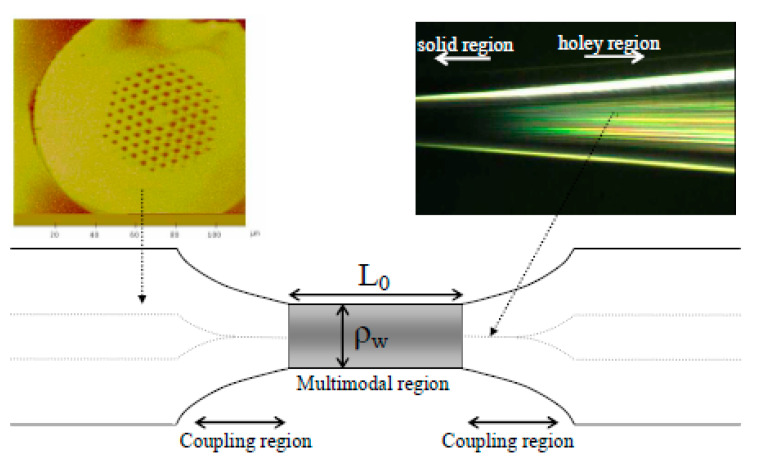
Images of the cross section of an SC-PCF (top left) used to fabricate the tapers and of the expanding zone of the taper (top right). The picture illustrates a tapered SC-PCF. The shadowed area represents the gas-permeable thin film. L_0_ is the length of the solid multimodal section and ρ_w_ is the taper waist diameter. Reprinted with permission from [77] ©2006 Optical Society of America.

**Figure 8 sensors-20-02555-f008:**
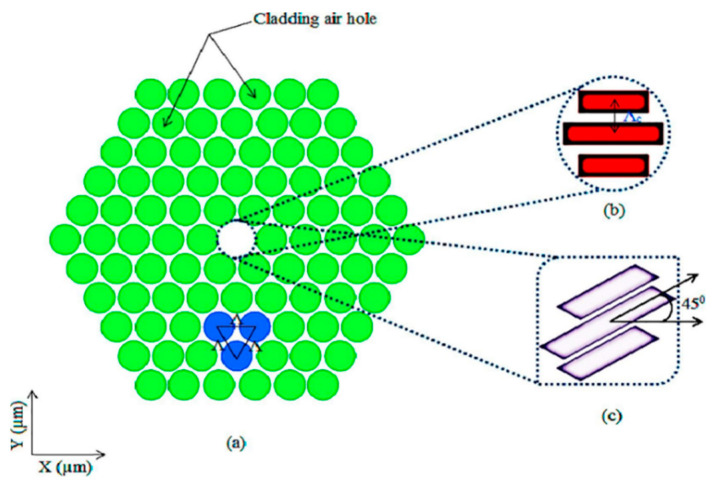
Schematic end face view of proposed PCF; (**a**) cladding region; (**b**) core region with 0° rotation; (**c**) core region with 45° rotation. Reprinted with permission from [84].

**Figure 9 sensors-20-02555-f009:**
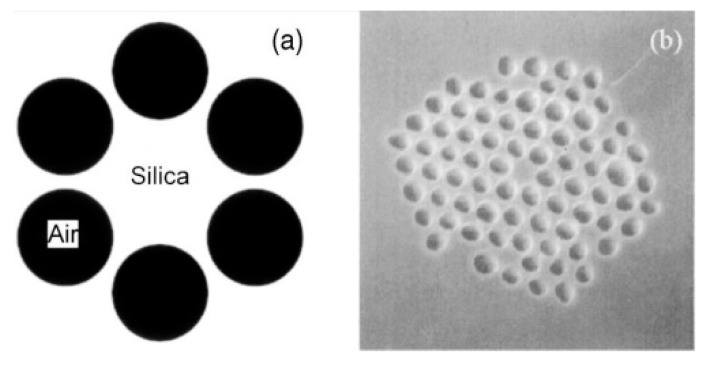
(**a**) Simulated MOF consisting of a 1.7 µm diameter core surrounded by a ring of 1.4 µm diameter air holes; (**b**) electron micrograph image of the inner cladding and core of the air–silica MOF. Reprinted with permission from [86] © 2000 Optical Society of America.

**Figure 10 sensors-20-02555-f010:**
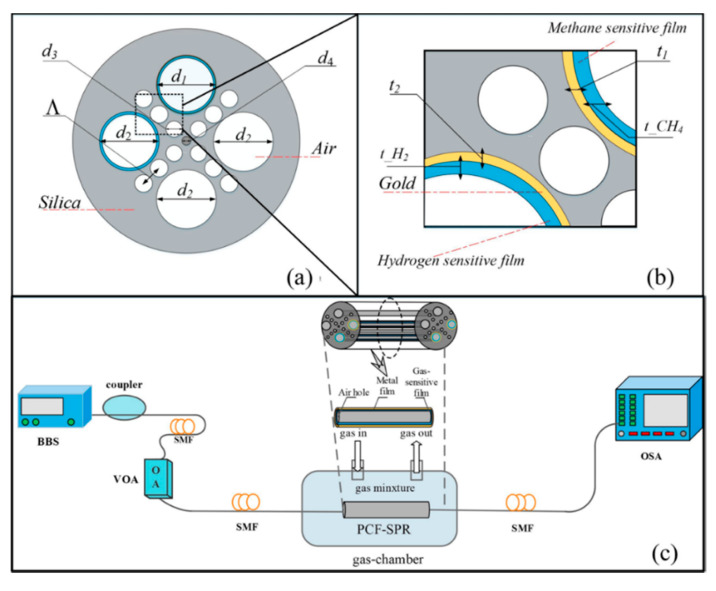
(**a**) The schematic and cross section of SC-PCF sensor; (**b**) structural parameters and; (**c**) experimental scheme. Reprinted with permission from [89].

**Figure 11 sensors-20-02555-f011:**
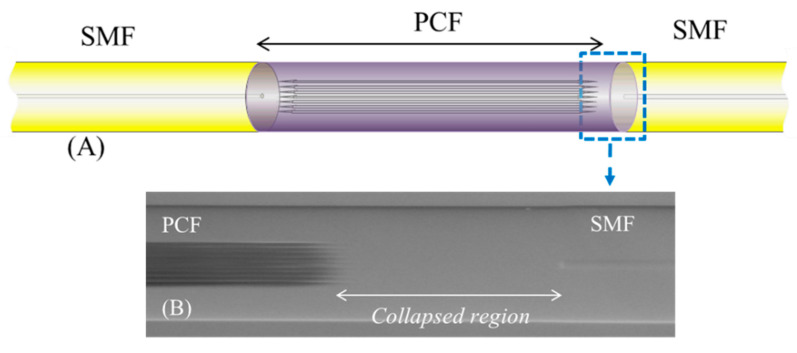
(**A**) Scheme of the splice between SC-PCF and SMF; (**B**) SEM photo of the PCF collapsed region.

**Figure 12 sensors-20-02555-f012:**
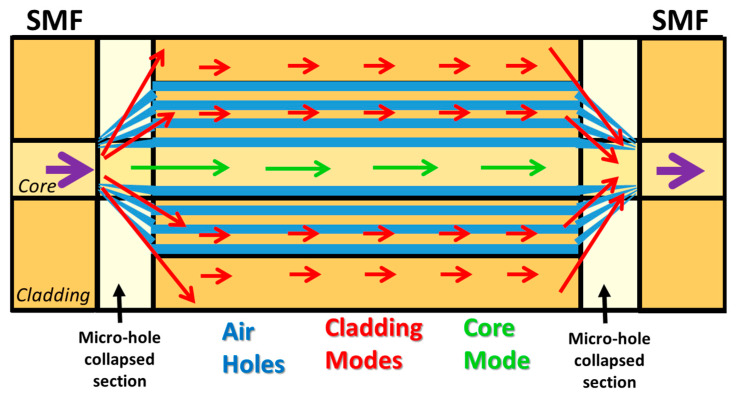
Operating mechanic of the Mach–Zenhder (MZ) interferometer based on SC-PCF. The arrows show the behavior of the cladding and core modes trough the SC-PCF; in blue are drawn the holes of the SC-PCF and in black the micro-hole collapsed sections.

**Figure 13 sensors-20-02555-f013:**
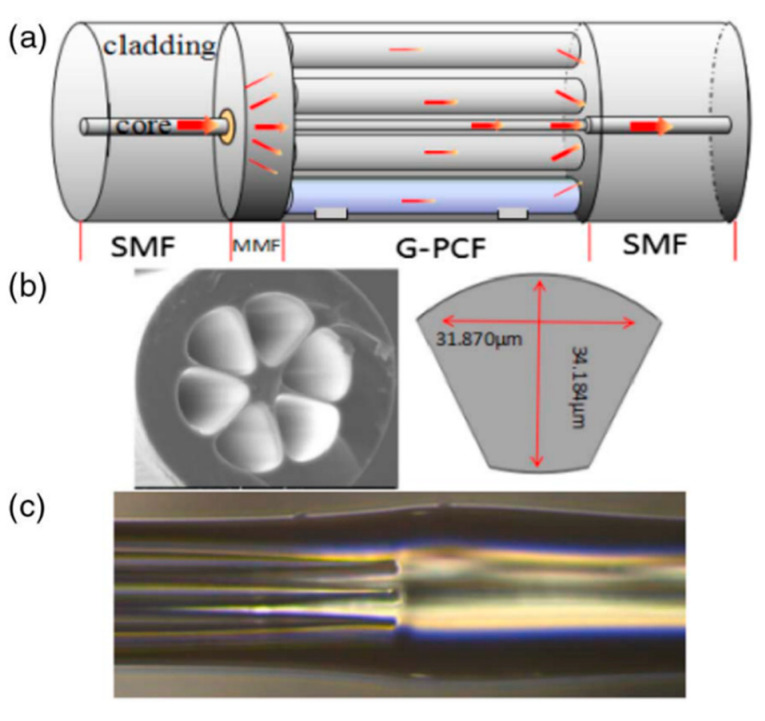
Schematic diagram of the MOF relative humidity (RH) sensor based on MZ interferometer; (**a**) cross section of the SC-PCF used; (**b**) optical microscopic image of the six-air-hole G-PCF cross-section structure and dimensions of the one of them; (**c**) zoom of the splice region between SC-PCF and SMF: the holes of the SC-PCF are completely collapsed. Reprinted with permission from [99] ©2018 Optical Society of America.

**Figure 14 sensors-20-02555-f014:**
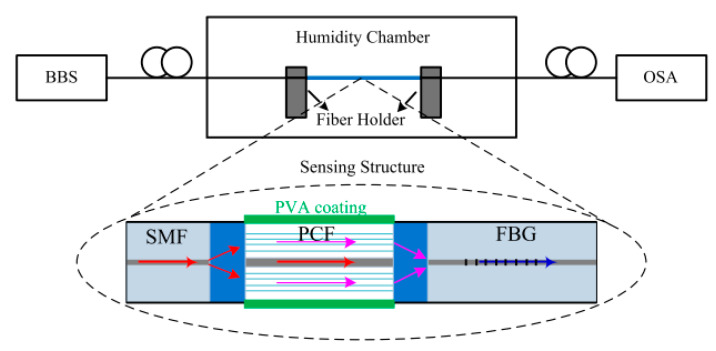
Schematic diagram of the experimental setup with the structure SMF-PCF-MZ interferometer and fiber bragg gratings (FBGs). Reprinted with permission from [100].

**Figure 15 sensors-20-02555-f015:**
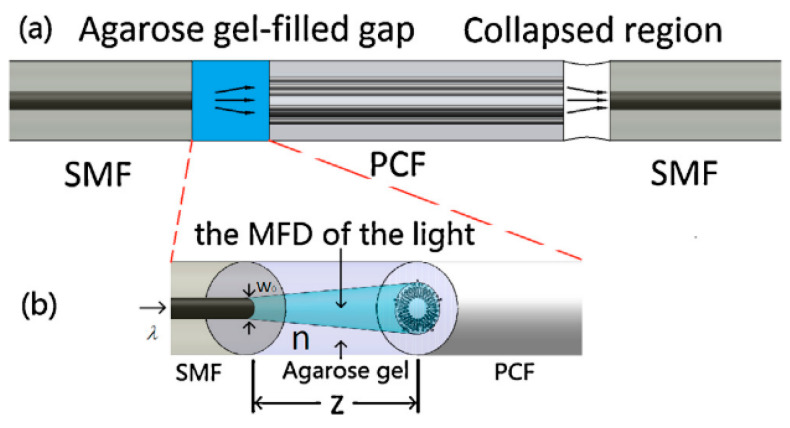
(**a**) Schematic diagram of the PCF based humidity sensor and; (**b**) the Mode Field Diameter (MFD) of the light in the Agarose gel. Reprinted with permission from [101].

**Figure 16 sensors-20-02555-f016:**
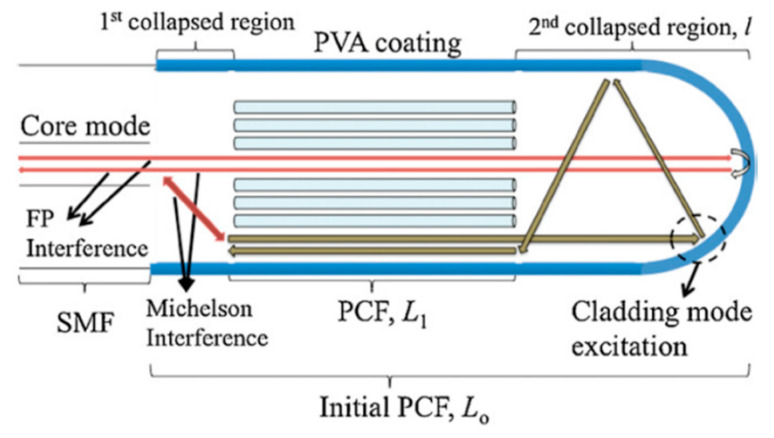
Schematic diagram showing the operating mechanism of the SC-PCF sensor. Reprinted with permission from [112].

**Figure 17 sensors-20-02555-f017:**
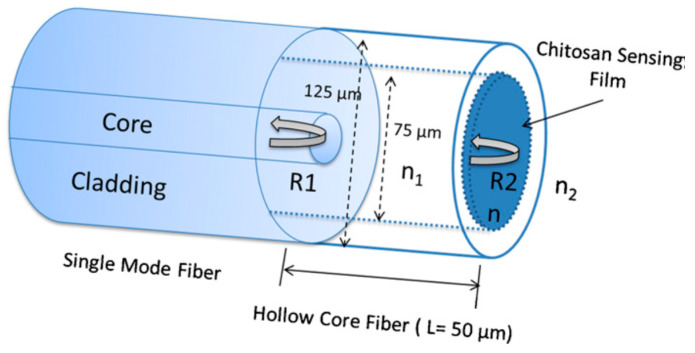
Structures of the proposed sensor based on a Fabry–Pèrot (FP) interferometer coated with chitosan film. Reprinted with permission from [117].

**Figure 18 sensors-20-02555-f018:**
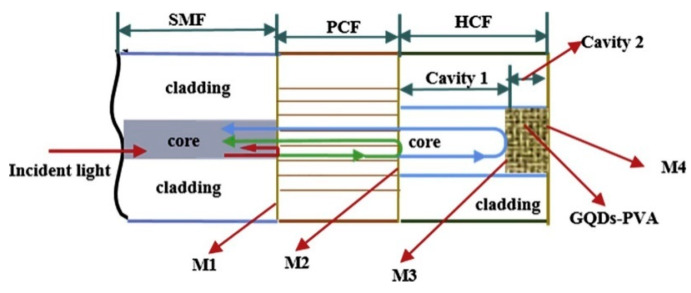
The structure of the FP interferometer constructed by splicing a section of an SC-PCF between a section of SMF and a section of HC-PCF. Reprinted with permission from [121].

**Figure 19 sensors-20-02555-f019:**
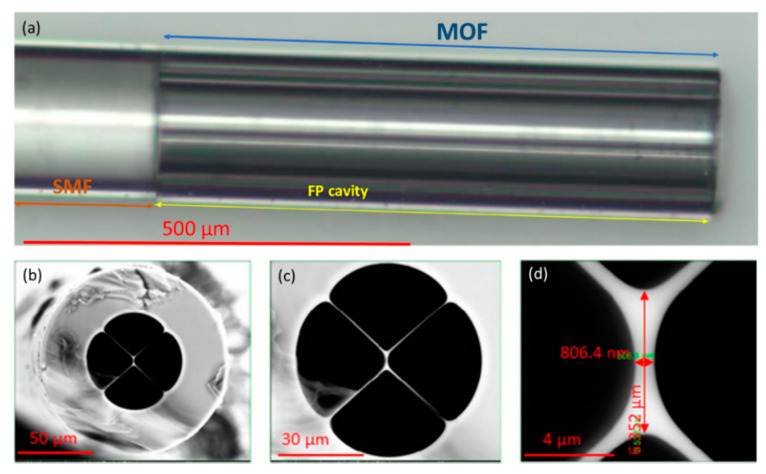
(**a**) SSC based on Fabry–Perot interferometer; (**b**) its cross-section and; (**c**) the four-bridge MOF core with (**d**) details of its geometry. Reprinted with permission from [123].

**Figure 20 sensors-20-02555-f020:**
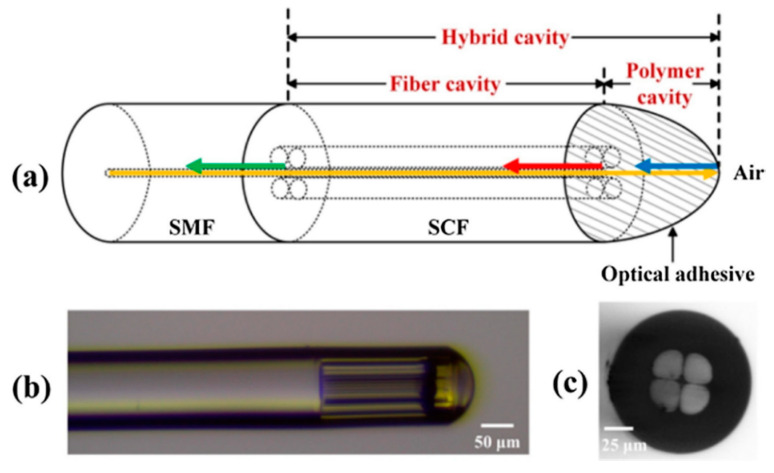
(**a**) Scheme of the different parts of the MOF sensor. The hybrid cavity is formed by the SSC cavity (SCF in the figure) and by the polymer cavity, formed by the OA; (**b**) Microscope image obtained of the developed MOF sensor; (**c**) Cross section of the 4-hole SSC. Reprinted with permission from [125].

**Figure 21 sensors-20-02555-f021:**
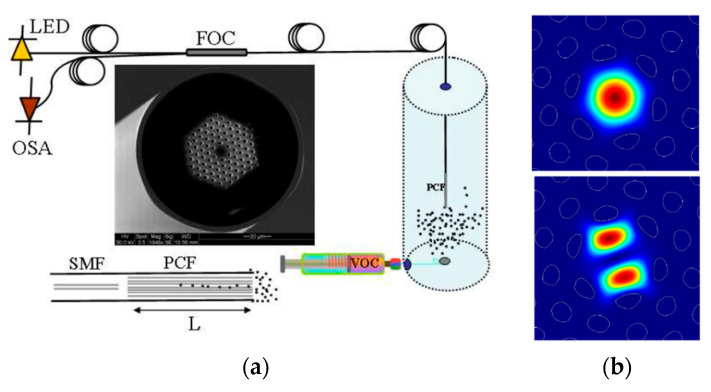
(**a**) Diagram of the experimental setup, micrograph of the SC-PCF used and drawing of the interferometer. L is the PCF length. LED stands for light emitting diode, FOC for fiber optic circulator or coupler, OSA for optical spectrum analyzer and SMF for single-mode fiber. The dots represent volatile organic compound (VOC) molecules; (**b**) transverse component of the electric field of the LP_01_- and LP_11_-like modes. Reprinted with permission from [130] ©2009 Optical Society of America.

**Figure 22 sensors-20-02555-f022:**
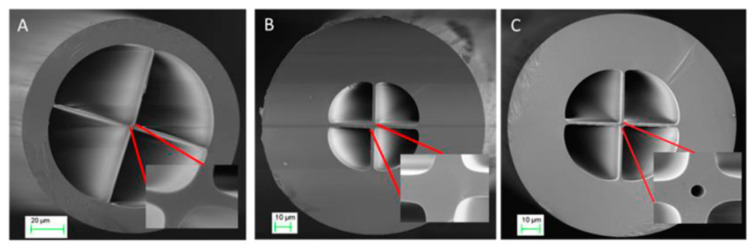
(**A**) SSC-1; (**B**) SSC-2; (**C**) SSC-3 cross section. In every figure, the detail of the core geometry can be appreciated and especially in (**C**), the hole located in the center of the optical fiber. Reprinted with permission from [135].

**Table 1 sensors-20-02555-t001:** Comparative table showing the most relevant features of MOF gas sensors listed in Section 3.

Reference	Gas	Sensitivity	LOD
[56]	C_2_H_2_	-	6 ppm
[57]	C_2_H_2_	-	2 ppb
[60]	C_2_H_2_	-	7ppm
[61]	C_2_H_2_	-	100 ppm (m/m)
[65]	C_2_H_2_	0.022dB/%	-
[68]	C_2_H_2_	-	0.05 % of saturated atmosphere
[69]	CH_4_	-	49 ppm (*v*/*v*)
[70]	CH_4_	-	647 ppm (*v*/*v*)
[71]	CH_4_	-	10 ppm (*v*/*v*)
[74]	CH_4_	-	below 0.1% of saturated atmosphere
[75]	H_2_	-	17 ppm (m/m)
[78]	N_2_	-	100 ppm (m/m)
[80]	C_2_ H_2_H_2_	-	5 ppm (m/m) 300 ppm (m/m)
[89]	CH_4_ H_2_	−1.99 nm/% −0.19 nm/%	- -
[90]	N_2_	−827.94 dB/RIU	-
[91]	NH_3_	-	8.09 ppt

**Table 2 sensors-20-02555-t002:** Table shown in [103] in which the features of the different AC-PCFI (Agarose Coated–Photonic Crystal Fiber Interferometer) devices at ~60%RH are exposed; the MOF sensor sensitivities had a high significant dependence on the agarose thicknesses. Reprinted with permission from [103] ©2013 Optical Society of America.

AC-PCFI	Average Red Peak Shift of the Coated PCFI Relative to the Uncoated PCFI (nm)	Refractive Index of Coating (Deduced from the RI Response of the PCFI)	Estimated Thickness of Coating (±100 nm) in nm
A	2.3	1.24	250
B	3.55	1.28	400
C	14.5	1.408	800
D	26.6	1.437	1250

**Table 3 sensors-20-02555-t003:** Comparative table showing the most relevant features of MOF humidity sensors listed in Section 4.

Reference	Sensitivity	Resolution	Range
[94]	-	0.2 %RH	0–90%RH
[95]	60.3 pm/%RH 188.3 pm/%RH	-	60–80%RH 80–85%RH
[96]	40.9 nm/%RH	-	20–95%RH
[97]	0.18 nm/%RH	-	30–85%RH
[98]	0.57 nm/%RH 1.43 nm/%RH	0.017 %RH 0.007 %RH	40–80%RH 80–95%RH
[99]	−0.077 dB/%RH	-	25–80%RH
[100]	-	0.13 %/RH	30–95%RH
[101]	2.2 dB/%RH	-	20–80%RH
[102]	0.0901 nm/%RH	-	13.77–77.87%RH
[103]	-	0.07 %RH	40–90%RH
[104]	-	0.074 %RH	20–95%RH
[105]	0.96 nm/%RH	0.067 %RH	20–90%RH
[106]	2.41 pm/%RH	-	27–85%RH
[107]	6.6 pm/%RH 24 pm/%RH	-	40–70%RH above 70%RH
[108]	0.14 dB/%RH	-	14–98%RH
[109]	0.093 dB/%RH	-	20–95 %RH
[111]	20.3 pm/%RH 61.6 pm/%RH	-	60–80 %RH 80–95 %RH
[113]	0.6 nm/%RH	-	30–90 %RH
[116]	0.058 nm/%RH	-	20–54 %RH
[117]	0.13 nm/%RH	-	20–95 %RH
[118]	0.083 nm/%RH	-	35–95 %RH
[119]	0.28 nm/%RH	-	at high RH% value
[120]	0.008 dB/%RH	-	5–90 %RH
[121]	0.456 nm/%RH	-	19,63–78,86 %RH
[122]	1.25 dB/cm	-	0–95 %RH
[123]	0.14 rad/%RH	0.0026 %RH	20–90 %RH
[125]	−0.042 read/%RH	-	15–90 %RH

**Table 4 sensors-20-02555-t004:** Comparative table showing the most relevant features of MOF VOC sensors listed in Section 5.

Reference	VOC	Sensitivity	LOD
[128]	C_2_H_5_Cl	-	30 ppb (*v*/*v*)
[130]	CH3CN THF	-	150 ppm 820 ppm
[131]	C_3_H_6_O	total shift of 1,3 rad	saturated atmospheres
[132]	C_2_H_5_OH	0.022 dB/ppm	-
[133]	C_2_H_5_OH	total shift 2.56 nm	250 ppm (m/m)
[134]	CH_3_OH	0.31π rad	saturated atmospheres
[135]	C_2_H_5_OH	close to 5π rad	saturated atmospheres

**Table 5 sensors-20-02555-t005:** Summary table of every sensor cited and explained in this review grouped by parameter detected and MOFs used.

	Absorption Lines			Effective Refractive Index Variations		
	SC-PCFs	HC-PCFs	SSCs	SC-PCFs	HC-PCFs	SSCs
**Gas sensors**	C_2_H_2_: [56,62,63,64,65,88] CH_4_: [56,88] H_2_: [77] O_2_: [79] H_2_N_2_: [84]	C_2_H_2_: [57,58,59,60,61,67,68,80,82,83,88] CH_4_: [69,70,71,72,73,74,83,88] H_2_: [75,76] N_2_: [78] CO: [80] H^13^CN: [82,83] NH_3_: [83]	C_2_H_2_: [66]	CH_4_: [89] H_2_: [89] N_2_: [90] NH_3_: [91]		
**Humidity sensors**		[93,94]		[95,96,97,98,99,100,101,102,103,104,105,106,107,108,109,110,111,112,113,114,115,116]	[117,118,119,120,121,122]	[123,124,125]
**VOCs sensors**		Aldehydes, ketones, hydrocarbons: [127]		C_2_H_5_Cl: [128] CH_3_CN and THF: [129,130] C_3_H_6_O: [131]	C_3_H_6_O: [131] C_2_H_5_OH: [132,133]	CH_3_OH: [134] C_2_H_5_OH: [135]

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
