# Peer review of "Optical Fiber Sensors Based on Microstructured Optical Fibers to Detect Gases and Volatile Organic Compounds—A Review"

_sensors, 2020, doi:10.3390/s20092555_

Round 1

Reviewer 1 Report

The manuscript presents a review on gasses detection using micro structured fibers that can be of interest for scientist and technicians working in the field.

The manuscript is placed in a proper context, contains representative number of references and the basic ideas of the previous work is reported in an accessible manner.

The manuscript must be amended as follows:

  1. Make clear in the introduction that the review includes only silica fibers (no polymer fibers are reported).
  2. Include magnetic sensors in the following sentence of the introduction:

“OFS can be used to measure physical or chemical magnitudes such as temperature, 29 curvature, displacement, pressure, refractive index, electric field, relative humidity (RH) and gasses,”

  1. Review and improve the figure captions:
  2. Describe the diagrams b, c and d in figure 6.
  3. Describe the picture and the diagram ‘b’ in figure 13. Furthermore, in the same figure, the part ‘b’ of the caption is incorrect, it refers to the picture ‘c’.
  4. Figure 16 is not a figure but a table, it must be referred as table 1.

Reviewer 2 Report

The manuscript provides an overview of the different approaches for the gas sensing employing microstructured optical fibers. It summarizes the results published in recent years. The manuscript is well written and can be interesting for the wide group of specialists. To the reviewer’s opinion, this manuscript meets requirements of MDPI Sensors. However, before publication, the following point should be addressed:

The Authors do not give an overview of the materials used for the fabrication of the microstructured optical fibers and provide the classification based on the geometry of the air holes only. For instance, plastic optical fibers (POF) and microstructured POF offer advantages over silica fibers such as low Young’s modulus, biological compatibility, high bending flexibility and ease of handling. These fibers operate in visible and near-IR light range.  There are dozens of papers and books related to the sensor applications based on the microstructured POFs, e.g.:

M. C. J. Large, L. Poladian, G. W. Barton, and M. A. van Eijkelenborg, Microstructured Polymer Optical Fibres. New York: Springer, 2007.

F. M. Cox, R. Lwin, M. C. J. Large, and C. M. B. Cordeiro, “Opening up optical fibers,” Opt. Exp., vol. 15, pp. 11843–11848, Sep. 2007.

A. Argyros, “Microstructured Polymer Optical Fibers,” J. Lightw. Technol., vol. 27, no. 11, 2009, pp.1571-1579.

J. Witt, M. Breithaupt, J. Erdmann, K. Krebber, „Humidity Sensing Based On Microstructured POF Long Period Gratings,” in Proc. Int. Conf. POF 2011, Bilbao, Spain, 2011, pp.409-414.

G. Woyessa, J.K.M. Pedersen, A. Fasano, et.al., “Zeonex-PMMA microstructured polymer optical FBGs for simultaneous humidity and temperature sensing,” Optics Letters, vol. 42, No. 6, 2017, pp. 1161-1164.

L.M. Pereira, A. Pospori, P. Antunes, et.al., “Phase-Shifted Bragg Grating Inscription in PMMA Microstructured POF Using 248-nm UV Radiation,” J. Lightw. Technol., vol. 35, no. 23, 2017, pp. 5176-5184.

Reviewer 3 Report

The authors presented a review of microstructured optical fibers (MOFs) to detect volatile organic compounds (VOC), relative humidity (RH) and gases. The classification of MOFs (suspended core fibers (SSCs) and photonic crystal fibers (PCFs)) and the transduction mechanisms (spectral absorption bands and variation of effective refractive index) to measure gas, RH and VOC were presented.

In my opinion, the paper needs several improvements in order to achieve the technical quality of Sensor MDPI Journal. There are many typos along the manuscript. The paper focus mainly in the physical structure of the sensors (fabrication of sensors arrange). However, I believe it is important addressing information about spectral characteristics.

Specifics comments:

  • In each sensing approach (Sections 3, 4 and 5), the authors could include a comparative table showing relevant information about each sensor such as resolution, sensitivity among others;
  • The authors must standardize the way the acronyms are introduced. For instance, “group velocity dispersion (GVD)” instead “Group Velocity Dispersion (GVD)”. In my opinion, the authors should not use capital letter at each word. Please, verify and correct it;
  • Page 2, line 64. The acronym is PCF instead PFC. Please verify and correct it;
  • Page 2, line 70. TIR was already defined;
  • Page 3, line 80. “Figure 3” instead “Figure 3(a)”;
  • The acronyms defined in the abstract should appear in the main text. Page 3, line 89. Please, define VOC;
  • Regarding Sections. The title must use capital letter at each word. For instance, “Transduction Mechanisms of Gas and VOC MOF Sensors”. Please, follow the MDPI template;
  • Regarding section 2, it seems very short to be a Section. In addition, it does not have references. Either improve it or merge it with other sections;
  • Page 5, line 142. “Yeuk L. Hoo et al.” instead “Yeuk L. Hoo et al”;
  • Regarding Figure 4. In the main text the authors claimed the fiber used for the reference gas cell was SC-PCF. However, in the Figure caption, the authors used HC-PCF. In addition, the acronyms used in the caption do not match with the Figure. Please, verify and correct it;
  • Page 6, line 192. “5 minutes” instead “5min”;
  • Regarding Figure 6. Please, describe in the Figure caption what means subfigures (a) to (c);
  • Page 13, line 448. Please, separate the values from the units. “3.3 pm/°C” instead “3.3pm/°C”;
  • Regarding Figures 11 and 12. The author must include the references in the Figures captions;
  • Regarding Figure 16. The authors should include the table instead the Figure. It looks weird;
  • Regarding caption of Figure 20(a). The acronym should be SSC-MOF instead SC-MOF? In addition, there is no reference in the figure;
  • Page 20, line 642. “Figure 22(a)” instead “Figure 22.a”. Please, verify and correct this typo along the manuscript;
  • There are several typos along the manuscript. Please carefully edit your manuscript by correcting typos, and please have a person fluent in English proofread your paper;

Round 2

Reviewer 3 Report

The authors have addressed correctly my suggestions. The manuscript can be accepted in its current form for publication in Sensors MDPI Journal.